# Evaluating model bias requires characterizing model mistakes

## Abstract

The ability to properly benchmark model performance in the face of spurious correlation is important to both build better predictors and increase confidence that models are operating as intended. We demonstrate that *characterizing* (as opposed to simply quantifying) model mistakes across subgroups is pivotal to properly reflect model biases, which are ignored by standard metrics such as worst-group accuracy or accuracy gap. Inspired by the well-established hypothesis testing framework, we introduce SkewSize, a principled and flexible metric that captures bias from mistakes in a model's predictions. It can be used in multi-class settings or generalised to the open vocabulary setting of generative models. SkewSize is an aggregation of the *effect size* of the interaction between two categorical variables: the independent variable, representing the bias attribute (i.e. subgroup), and the dependent variable, representing the model's prediction. We demonstrate the utility of SkewSize in multiple settings including: standard vision models trained on synthetic data, vision models trained on ImageNet as well as the DomainNet distribution shift benchmark, and large scale vision-and-language models from the Blip-2 family. In each case, the proposed SkewSize is able to highlight biases not captured by other metrics, while also providing insights on the impact of recently proposed techniques, such as instruction tuning.

## 1 Introduction

Machine learning systems can capture unintended biases (Dixon et al., 2018) by relying on correlations in their training data that may be spurious (i.e. a finite sample artifact), undesirable and/or that might vary across environments. Models of all scales are vulnerable to this failure mode, including recent, large-scale models (Weidinger et al., 2022; Birhane et al., 2023; Luccioni et al., 2023; Solaiman et al., 2023). To evaluate unintended biases in model outputs, existing metrics divide the population (or test set) into subgroups (based on demographic characteristics, how well-represented in the dataset each group is, or another characteristic of significance) and aggregate the e.g. correct and incorrect outputs across those subgroups as in Sagawa et al. (2019). However, existing metrics consider as equivalent all responses deemed to be incorrect, obscuring important information regarding a model's bias characteristics, especially in the context of large or intractable output spaces.

**Motivating example.** Consider the synthetic setup in Figure 1 which compares two image classification models: Model 1 and Model 2. These models predict occupation, with different distributions of outputs across two mutually exclusive[1] subgroups (male and female). Following prior work, we first compute model accuracy in each subgroup (e.g. Chowdhery et al., 2022), worst group accuracy (i.e. minimum accuracy across groups, Sagawa et al., 2019) and Gap (the difference between subgroup accuracy and overall accuracy, Zhang & Ré, 2022):

*Case 1* (Writer). Model 2's accuracy is lower than that of Model 1; a bias in Model 2's predictions is evident in women being misclassified as Editors and men being misclassified as Composers and Philosophers. Accuracy and Worst group accuracy degrade as expected for the more biased model, whereas Gap does not.

---

[1]Assumed to be mutually exclusive for the limited purpose of this illustrative example. We recognize that reality is richer and more nuanced than this binary categorization. See *Limitations* section.

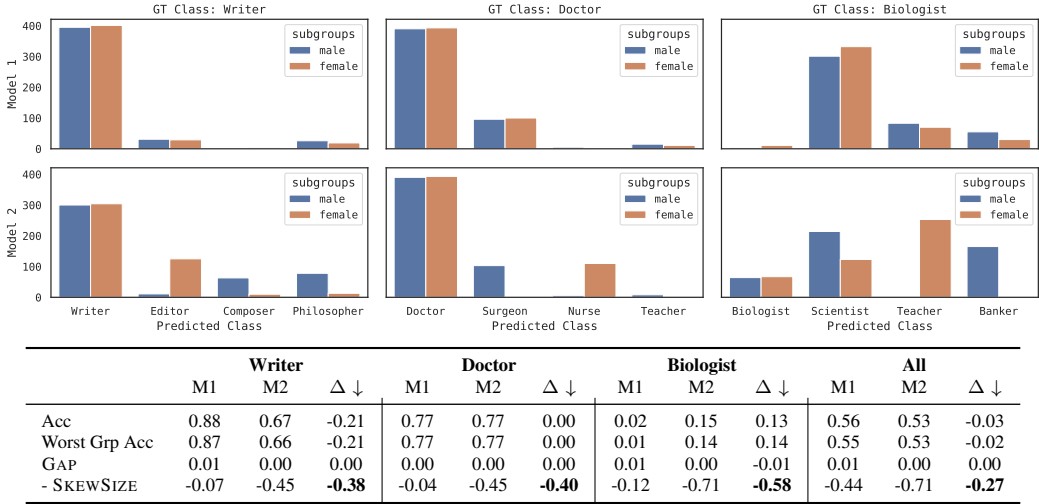

| | Writer | | | Doctor | | | Biologist | | | All | | |
|---|---|---|---|---|---|---|---|---|---|---|---|---|
| | M1 | M2 | Δ↓ | M1 | M2 | Δ↓ | M1 | M2 | Δ↓ | M1 | M2 | Δ↓ |
| Acc | 0.88 | 0.67 | -0.21 | 0.77 | 0.77 | 0.00 | 0.02 | 0.15 | 0.13 | 0.56 | 0.53 | -0.03 |
| Worst Grp Acc | 0.87 | 0.66 | -0.21 | 0.77 | 0.77 | 0.00 | 0.01 | 0.14 | 0.14 | 0.55 | 0.53 | -0.02 |
| GAP | 0.01 | 0.00 | 0.00 | 0.00 | 0.00 | 0.00 | 0.01 | 0.00 | -0.01 | 0.01 | 0.00 | 0.00 |
| - SKEWSIZE | -0.07 | -0.45 | **-0.38** | -0.04 | -0.45 | **-0.40** | -0.12 | -0.71 | **-0.58** | -0.44 | -0.71 | **-0.27** |

Figure 1: **Motivating example: standard metrics fail to capture subtle biases within a model.** We plot the prediction counts for two models given three ground-truth classes (WRITER, DOCTOR, BIOLOGIST). MODEL 1 (M1) displays similar distributions of errors for both subgroups whereas MODEL 2 (M2) displays 'stereotypical' errors (e.g. mispredicting female DOCTORs for NURSEs). In the table, we report accuracy (Acc), worst group accuracy (Worst Grp Acc), GAP and their difference (Δ) between M1 and M2. Only our approach (SKEWSIZE) captures the bias in all settings.

*Case 2* (DOCTOR). Accuracy is the same for MODEL 1 and MODEL 2 but a bias is evident in MODEL 2's predictions, with women being misclassified as NURSEs, and men being misclassified as SURGEONs. Traditional metrics do not capture this bias.

*Case 3* (BIOLOGIST). Accuracy is higher for MODEL 2 than MODEL 1, but a bias is evident in MODEL 2's predictions, with women being misclassified as TEACHERs and men being misclassified as SCIENTISTs or BANKERs. Counterintuitively, the standard metrics improve or stay the same.

*All.* Aggregating across classes, we can see that the standard metrics either improve in MODEL 2 relative to MODEL 1 or do not change.

In light of this example, we see that regardless of how the performance of a model *in terms of accuracy* varies across subgroups, bias may also arise from systematic errors in the incorrect predictions made. Importantly, previously proposed metrics do not surface such bias and give the misguided impression that the model's predictions do not exhibit bias. To measure this type of bias, we introduce SKEWSIZE, which considers *how different* the distribution of predictions are across subgroups. In our motivating example, SKEWSIZE is able to capture the different types of biases.

We formulate the problem of estimating bias by viewing it through the lens of hypothesis testing. We draw inspiration from tests of association between the confounding, spurious factor (e.g. gender) and the model's (discrete) prediction; in particular, we re-purpose a common measure of *effect size* for such tests. We compute effect sizes of this association for each ground-truth class: for instance, given images of doctors, we can estimate the effect size corresponding to the relationship between gender and predicted occupation. Finally, we aggregate these effect sizes across a set of classes using a measure of the skewness of the effect size distribution to arrive at a scalar metric which can be used to compare different models.

We investigate the utility of this metric in three settings. *(1) Synthetic data:* First, we investigate the utility of the metric, relative to existing metrics, on a synthetic task, and demonstrate that our metric captures bias not reliably captured by other metrics. *(2) Distribution shift / multi-class classification:* Next, we demonstrate how our metric can be used to identify previously unidentified cases of systematic bias arising from model mispredictions using DOMAINNET and IMAGENET (Deng et al., 2009; Peng et al., 2019). *(3) Open-ended prediction:* Finally, we analyse large scale vision-

and-language models (BLIP-2, Li et al. (2023)) that have an intractable[2] output space using our metric in two separate settings: gender vs. occupation and gender vs. practiced sport. In settings (2) and (3), we find that no datasets exist that would allow for computing bias metrics among subgroups with statistical significance. As a result, for these settings, we create synthetic datasets in order to run our evaluation at scale. Our main contributions can be summarized as follows:

1. We demonstrate limitations of current metrics for quantifying bias, specifically that they fail to capture bias manifested in *how the model makes mistakes*.
2. We propose SKEWSIZE, a metric for evaluating bias in discriminative models inspired by hypothesis tests of contingency tables.
3. We use SKEWSIZE to robustly evaluate model bias at scale in a variety of domains and identify systematic biases arising in the models' errors. We further show how SKEWSIZE can be used with synthetic data to evaluate gender bias in vision-and-language models (VLMs) on two tasks.

## 2   METHOD

### 2.1   BACKGROUND

**Notation.** We consider a discriminative model $f_\theta : \mathcal{X} \mapsto \mathcal{Y}$ with parameters $\theta$, where $\mathcal{X}$ is the set of inputs (e.g. images) and $\mathcal{Y}$ is the label set. We also assume that input $x \in \mathcal{X}$ with label $y \in \mathcal{Y}$ is drawn from an underlying distribution $p(x|z, y)$, where $z$ is a discrete latent variable $z \in \mathcal{Z}$ that represents a factor of variation that affects the data generating process. In the context of this work, $z$ is referred to as the *bias* variable and assumed to systematically affect how well the model $f_\theta$ is able to predict $y$ from $x$. Our goal is then estimating to what extent the predictions are affected by $z$.

**Metrics for output disparity across subgroups.** Previous work on evaluating performance disparity across subgroups has mostly considered metrics such as accuracy (Zhang & Ré, 2022; Alvi et al., 2018; Li et al., 2022), worst group accuracy (Zhang & Ré, 2022; Koh et al., 2021), gap between average and worst group accuracy (referred to as GAP, Zhang & Ré, 2022). These metrics focus on the true positive rate and do not identify biases in the distribution of prediction errors. We compute these metrics throughout the work, for comparison with our approach.

Alternatively, fairness criteria can be formulated as independence desiderata (Barocas et al., 2019), with metrics classified as 'independence' criteria if $f_\theta(x) \perp z$, 'separation' if $f_\theta(x) \perp z|y$ and 'sufficiency' if $y \perp z|f_\theta(x)$. In practice, these criteria are operationalized using different metrics. For the independence criterion, demographic parity (Dwork et al., 2012, DP) is commonly used. These metrics have been recently extended for use in the multiclass setup (e.g. Alabdulmohsin et al., 2022; Pagano et al., 2023; Putzel & Lee, 2022; Rouzot et al., 2022). In this case, metrics are typically computed by binarizing each class (e.g. Alabdulmohsin et al., 2022; Pagano et al., 2023) and aggregating fairness scores across classes using their maximum (i.e. worst case scenario), or average (c.f. Appendix A). Given a full confusion matrix, equality of odds (EO) (Hardt et al., 2016), and potentially DP, would capture differences in the distributions of model errors. However, the detected bias would be surfaced in the scores of the confused classes rather than associated with the class of interest. In our motivating example, EO comparing MALE and FEMALE examples in the DOCTOR class would be close to 0, but larger for the SURGEON and NURSE classes. In an intractable output space, a full confusion matrix may be unavailable, and EO and DP would be limited in their ability to highlight differences in the distribution of model errors. In this work, we compute EO and DP as per Alabdulmohsin et al. (2022) when a full confusion matrix is available.

### 2.2   ESTIMATING DISTRIBUTIONAL BIAS FOR CATEGORICAL DISTRIBUTIONS

Let $\mathcal{Z} = \{A, B\}$ and $\mathcal{Y}$ be a discrete set. We further consider that the parametric model $f_\theta(x)$ defines a conditional distribution $q(y|x; \theta)$ for each $x \in \mathcal{X}$. For a fixed value of $y' \in \mathcal{Y}$, distributional bias should account for systematic differences in the outcomes of $f_\theta(x)$ across different subgroups, i.e. when $x$ is sampled from $p(x|y, z = A)$ versus $p(x|y, z = B)$. More formally, in Equation 1, we define distributional bias as a comparison between *induced families of distributions* defined by $f_\theta(x)$ when $x \sim p(x|y = y', z = A)$ versus when $x \sim p(x|y = y', z = B)$:

$$\mathcal{H}(Q_A(y|x; \theta)||Q_B(y|x; \theta)), \tag{1}$$

---

[2]This refers to the setting where the label space is given by all the possible outputs of a language model.

where $Q_A(y|x;\theta)$ and $Q_B(y|x;\theta)$ denote the family of distributions obtained when the bias variable assumes each of its possible values, i.e. $z = A$ and $z = B$, respectively. $\mathcal{H}(\cdot||\cdot)$ represents an operator that accounts for a notion of similarity between the two distributions. Depending on the nature of $Q$, $\mathcal{H}$ can assume different forms. Also, notice that $\mathcal{H}$ operator is not limited to binary attributes and can be instantiated by approaches to compare multiple families of distributions.

As we focus on classification tasks, $f_\theta(x)$ parameterizes families of categorical distributions. We can thus formulate the comparison between $Q_A$ and $Q_B$ as *estimating the effect size* of the association between the bias variable $z$ and the observed model predictions $y' \sim q(y|x,z)$. In this framework, the measure of similarity between $Q_A$ and $Q_B$ can be seen as a measure of association between two categorical variables, the independent variable representing the bias attribute $z$ and $y'$, which we propose to be estimated as per the Cramér's V statistic (Cramér, 1946), and is defined as:

$$\nu = \sqrt{\frac{\chi^2}{N \cdot DF}}, \tag{2}$$

where $N$ the sample size $DF$ is the number of degrees of freedom, and $\chi^2$ represents the test statistic from the corresponding Pearson's chi-squared independence test. Cramér's V is bounded between 0 and 1, with 1 indicating a perfect association between both variables, i.e. the predictions are extremely sensitive to changes in the value of the bias variable. In order to compute the value of $\chi^2$, the counts of predictions must be arranged in a *contingency table* of size $M = |\mathcal{Z}| \cdot |\mathcal{Y}|$. For a given class $y'$, each entry should correspond to the frequency with each predicted class was observed per subgroup in the data. The $\chi^2$ statistic for such observations accounts for the discrepancy between observed and expected frequencies in the table and is defined as:

$$\chi^2 = \sum_{k=1}^{M} \frac{(o_k - e_k)^2}{e_k}. \tag{3}$$

where $o_k$ refers to the observed value of the $k$-th entry in the contingency table, and $e_k$ refers to the expected value of this table entry under the assumption of independence between the bias variable and the prediction.

## 2.3 AGGREGATING THE EFFECT SIZE

The effect size based approach to measure distributional bias evaluates model predictions on a per-class basis. In order to obtain a single, scalar summary metric which can be used to compare multiple models, we must consider how to aggregate the estimated effect sizes for all classes. The ideal metric should be able to simultaneously satisfy the following two conditions: (i) indicate an overall weaker bias when the distribution of effect size values per class is centered around zero with infrequent higher values (as classes for which the model is strongly affected by bias are rare), (ii) distinguish models weakly exhibiting bias from models where, for a considerable fraction of classes, the predictions exhibit high degrees of association with the bias variable, (i.e., the distribution of effect size values is long-tailed and skewed towards the left).

Given the aforementioned desiderata, we propose to aggregate the effect size values per class using the Fisher-Pearson coefficient of skewness, as it captures both how *asymmetric* the distribution of estimated effect size values is as well as the *direction* of the asymmetry. For estimated effect sizes $\{\nu_1, \nu_2, \ldots, \nu_{|\mathcal{Y}|}\}$ with empirical mean $\bar{\nu}$, the proposed metric SKEWSIZE is computed as:

$$\text{SKEWSIZE} = \frac{\sum_{i=1}^{|\mathcal{Y}|} (\nu_i - \bar{\nu})^3}{\left[\sum_{i=1}^{|\mathcal{Y}|} (\nu_i - \bar{\nu})^2\right]^{3/2}}. \tag{4}$$

In the Appendix we provide pseudocode for SKEWSIZE (Algorithm 1), a Python implementation, and a discussion on modulating the impact low count predictions have when computing SKEWSIZE.

**Flexibility of SKEWSIZE.** As we are interested in surfacing biases in the distribution of model errors, our formulation can be related to a 'separation' fairness criterion. However, this does not preclude the implementation of other criteria. For instance, sufficiency could be implemented by conditioning on the predicted class and using the ground-truth class as the independent variable in the $\chi^2$ test. Similarly, we can implement DP by using the model outputs as the independent variable. Finally, SKEWSIZE can be computed based on logits, softmax scores, top-1 or top-$k$ predictions. Here, we focus on the separation formulation based on top-1 predictions for simplicity.

| | Accuracy-based | | | DP($\downarrow$) | | | EO ($\downarrow$) | | | **Effect size** ($\downarrow$) | | |
|---|---|---|---|---|---|---|---|---|---|---|---|---|
| Removed | Accuracy ($\uparrow$) | WG ($\uparrow$) | GAP ($\downarrow$) | Class 0 | Class 1 | Class 2 | Class 0 | Class 1 | Class 2 | Class 0 | Class 1 | Class 2 |
| Unbiased | 0.998 | 0.996 | 0.002 | 0.004 | 0.004 | 0.001 | 0.001 | 0.001 | 0.001 | 0.006 | 0.020 | 0.024 |
| Class 0 | 0.888 | 0.666 | 0.222 | 0.050 | 0.315 | 0.265 | 0.038 | 0.483 | 0.203 | 0.705 | 0.012 | 0.017 |
| Class 1 | 0.891 | 0.653 | 0.238 | 0.303 | 0.013 | 0.289 | 0.448 | 0.009 | 0.220 | 0.012 | 0.703 | 0.011 |
| Class 2 | 0.888 | 0.664 | 0.224 | 0.278 | 0.057 | 0.332 | 0.208 | 0.040 | 0.484 | 0.032 | 0.009 | 0.700 |

Table 1: DSPRITES. ResNet18s trained on 4 versions of DSPRITES. Effect size is only non-negligible for the corresponding biased classes.

## 3 EXPERIMENTS

In this section, we first empirically demonstrate the usefulness of SKEWSIZE to evaluate models trained to perform tasks of increasing difficulty on three datasets: dSprites (Matthey et al., 2017), DOMAINNET (Peng et al., 2019), and IMAGENET (Deng et al., 2009). We follow with an application of SKEWSIZE for assessing VLMs from the BLIP-2 family and demonstrate it can be used in cases where predicted classes do not necessarily appear as ground-truth in the evaluation dataset.

### 3.1 A PROOF-OF-CONCEPT: DSPRITES DATASET

We begin with a simplified setting using the DSPRITES dataset, which contains images with objects represented by different shapes, colors and at different positions. We leverage knowledge about the data generating process to introduce biases in the obtained model by excluding training examples of a specific type. This allows us to validate whether effect size estimation can be used as a strategy to capture biased predictions by investigating the hypothesis that a non-negligible effect size would only be observed for class, subgroup pairings that were not seen at training time.

**Setting.** We consider the task of predicting the object *shape* and evaluate whether the model predictions are biased with respect to object *color* under a regime of systematic training set manipulation. Using the terminology in Section 2, the object color is the *independent variable* (i.e. the variable on which we intervene), and the predicted shape is the *dependent variable* (i.e. the variable we observe).

**Data and Models.** Given the DSPRITES' label set {Heart, Ellipsis, Square} and the set of values for the color attribute {Red, Green, Blue}, we build three different versions of the training data, where for each version we remove all examples with GREEN color from one of the classes. We then train a ResNet18 (He et al., 2016) for 5,000 steps with each dataset and evaluate on held out data that has *not* been manipulated to remove any GREEN examples. For each ground-truth class, we compute the effect size of the interaction between color and prediction as described in Section 2. For completeness, we verify that each model reaches nearly 100% accuracy on a test set biased in a similar way. These experiments highlight the existence of bias in the model's predictions: accuracy should be nearly 100% for all classes but the removed one, in which case errors mostly occur when the color of the object is GREEN.

**Results.** We present in Table 1 results for the three models and one with the same architecture but trained with unbiased data. We report accuracy on an *unbiased* test set, along with worst group accuracy, GAP, EO, DP, and per-class effect size (our approach). As intended, models trained with biased data had lower accuracy in comparison with the model trained with unbiased data, suggesting that models trained with biased data indeed came to rely on the spurious correlation to some extent. We observe that, for all models, effect sizes were strong only for the classes that had green instances removed at training time, while remaining negligible for the other classes, confirming the hypothesis that the proposed approach indeed captures model biases and can provide per-class granularity. In contrast, EO and DP tend to distribute the effect of this bias across the confused classes, and do not readily indicate the origins of the confusion.

### 3.2 ESTIMATING DISTRIBUTIONAL BIAS IN MULTI-CLASS CLASSIFICATION: DOMAINNET

We now stress-test SKEWSIZE by employing it to evaluate a model in the multi-domain setting, where samples from different distributions are employed training time, and show that our proposed metric can capture systematic biases in predictions. Specifically, we investigate the degree of bias

exhibited by the model with respect to the different domains (in this setting, the domain label corresponds to the spurious bias variable).

**Setting.** We consider the DOMAINNET benchmark (Peng et al., 2019), which is composed of images from 6 domains sharing the same label space of 345 object classes, and train a ResNet-50 on the train split of all domains jointly. Given the trained model, we then compute predictions for all instances in the test partitions and proceed to compute SKEWSIZE as per Algorithm 1.

**Results.** The model achieved 59.95% average test accuracy, 37.01% worst group accuracy gap, and 0.509 SKEWSIZE. In order to provide a fine-grained understanding about the differences between each metric, we show in Figure 2 plots accuracy (per class) against effect size $\nu$, along with the respective Equality of Odds (EO) value (shown as each point's corresponding hue). We find a mild Pearson correlation between effect size and accuracy ($-0.291$, $p \approx 0$) as well as between effect size and EO ($0.190$, $p = 0.0008$), which indicate the metrics are related but not equivalent as they capture distinct aspects of the bias. No correlation between effect size and GAP was found ($0.103$, $p = 0.07$), nor between effect size and DP ($0.051$, $p = 0.377$) further highlighting the importance of including robustness evaluations metrics that take into account error mismatches for a given ground-truth class.

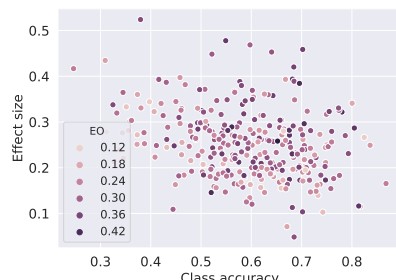

Figure 2: DOMAINNET. Per-class accuracy vs. effect size. Hue indicates EO. Points in the top-right most corner of the plot indicate that even for classes where the model is most accurate systematic differences in predictions across subgroups might exist.

### 3.3 ESTIMATING DISTRIBUTIONAL BIAS IN MULTI-CLASS CLASSIFICATION: IMAGENET

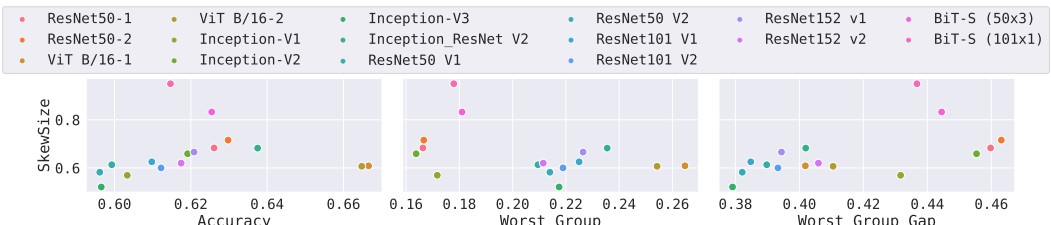

Figure 3: IMAGENET. We plot SKEWSIZE versus each accuracy based metric for a variety of models. No accuracy based metric presented a clear trend with respect to SKEWSIZE, demonstrating that the proposed metric identifies issues with models not exposed by these other metrics.

We have thus far demonstrated that SKEWSIZE is capable of accounting for aspects of a model's behaviour that are not captured by accuracy-based bias metrics. We now further examine the motivating example presented in Section 2 and showcase how SKEWSIZE can be used to capture different bias levels even when models perform similarly in terms of accuracy.

**Models.** We consider models spanning four architectures: RESNET50S (He et al., 2016), VISION TRANSFORMERS (VITS) (Dosovitskiy et al., 2020), INCEPTION (Szegedy et al., 2015), and BIT models (Kolesnikov et al., 2020). Architecture and training details are described in Appendix E.

**Data.** In order to evaluate the SKEWSIZE of each model, we consider a scenario where the background of an image corresponds to the bias variable. As no background annotations are available in the original IMAGENET, we chose 200 classes from the original label set (specifically, those present in TINYIMAGENET (Le & Yang, 2015)) and generated a synthetic dataset containing images of each of the selected classes across 23 different background types (list obtained from Vendrow et al. (2023)) using STABLE DIFFUSION (Rombach et al., 2022). We generate images using the prompt template *A photo of a* {CLASS} {BACKGROUND} . For instance, for the class SALAMANDER, we

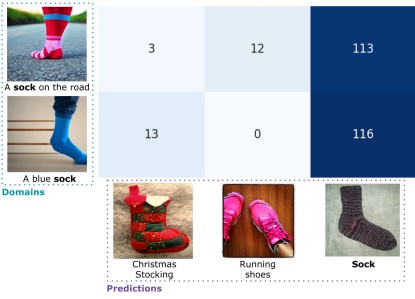

Figure 4: **Bias exposed by SKEWSIZE.** Both domains for the SOCKS class have similar accuracy, but the mismatch in errors suggests the model relies on spurious correlations based on background/color.

| Occupation | Accuracy (↑) | GAP (↓) | Effect size (↓) |
|---|---|---|---|
| Writer | 80.24% | 0.60% | 0.263 |
| Doctor | 90.32% | 7.26% | 0.291 |
| Biologist | 1.51% | 0.71% | 0.250 |
| Maid | 31.75% | 12.00% | 0.556 |
| Model | 83.77% | 10.18% | 0.368 |
| Nurse | 51.71% | 35.79% | 0.728 |
| Philosopher | 34.88% | 34.68% | 0.927 |
| Scientist | 73.69% | 6.55% | 0.241 |
| Veterinarian | 79.13% | 0.10% | 0.154 |

Table 2: **VLM evaluation.** Even in cases where the accuracy gap is nearly 0, results show there still is a significant interaction between gender and predicted occupations (c.f. the writer class), indicating the existence of gender bias which is not reflected by metrics based on accuracy.

generate images using *A photo of a* SALAMANDER ON THE ROCKS . We generate 200 images for each background-class pair. Note that these images are used only for evaluation, not training.

**Results.** In Figure 3 we compare models in terms of accuracy, worst group accuracy, worst group accuracy gap, and SKEWSIZE. We observe no clear correlation between these metrics and SKEWSIZE: models with similar accuracy may have significant bias in their prediction error distributions and vice versa, demonstrating SKEWSIZE provides complementary information to other metrics.

**Qualitative examples.** We now more closely examine specific examples of systematic bias uncovered by SKEWSIZE. We identify examples by picking classes where the model is both accurate and the effect size for the association between subgroup (background) and the model's prediction is high. In Figure 4, we show the top-3 predictions by the VIT B/16-1 for SOCKS in subgroups corresponding to *A photo of a* SOCK ON THE ROAD and *A photo of a* BLUE SOCK . Despite similar measured accuracy, the differences in the most frequent errors for each subgroup captured by SKEWSIZE suggest that the evaluated model may incorrectly associate an ON THE ROAD background with the class RUNNING SHOES, even when the true object of interest is absent.

## 3.4 COMPARING VLMS FOR MULTI-CLASS CLASSIFICATION ACROSS MODEL SIZE

We now consider the case where the output space is intractable and obtaining data to evaluate the model is challenging. We study the BLIP-2 model family for (binary) gender bias when predicting occupation or practiced sport.

**Data.** Apart from Visogender (Hall et al., 2023b), with only 500 instances, there are no real-world datasets available for evaluating gender biases on VLMs. Therefore, to investigate the utility of SKEWSIZE in the evaluation of VLMs, we gather synthetic data with templates constructed as follows. For the occupation task, we use templates of the form *A* {GENDER} {OCCUPATION}. and query the VLM model with *What is this person's occupation?* . In order to evaluate the model under conditions that most closely resemble their usage "in-the-wild", we *directly* use the textual output as predicted class and *do not* constrain the output space of the VLM in order to obtain predictions within the label set of the generated dataset. More details are given in Appendix D.4.

**Models.** We again leverage STABLE DIFFUSION(Rombach et al., 2022) to generate data in order to investigate models from the BLIP-2 family with different characteristics (size, instruction tuning). Specifically, we consider BLIP-2 ViT-g OPT$_{2.7B}$ (BLIP2-2.7B) with 3.8B parameters and an unsupervised-trained language model, BLIP-2 ViT-g OPT$_{6.7B}$ (BLIP2-6.7B), with 7.8B parameters and an unsupervised-trained language model, and BLIP-2 ViT-g FlanT5$_{XL}$ (BLIP2-FlanT5), with 4.1B parameters and an instruction-tuned language model.

**Results.** We report effect size for various occupations in Table 2 considering predictions by BLIP2-FlanT5. By comparing the accuracy and GAP with effect size for the three classes reported in

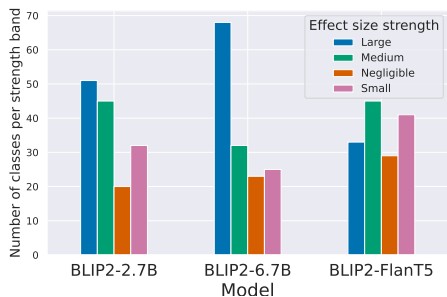

(a) Gender bias in occupation prediction.

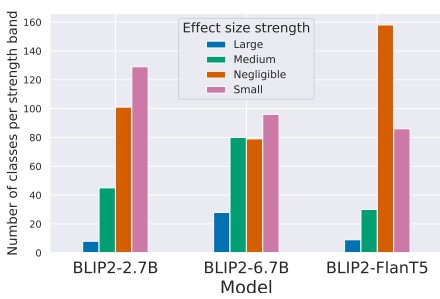

(b) Gender bias in sport modality prediction.

Figure 5: **Effect size comparison - BLIP-2 models.** We categorize effect size values in strength bands: 0-0.1 is a negligible effect, while 0.1-0.3, 0.3-0.5, and above 0.5 correspond to small, medium, and large, respectively. Scaling up model size with an unsupervised-trained language model amplified the bias, whereas instruction-tuning the language model seemed to mitigate it.

Table 1, namely WRITER, DOCTOR, and BIOLOGIST, we further validate the main premise of this work. Results for the remaining classes provide evidence that when GAP is high, the effect size also increases, further showcasing the potential of such a metric to measure disparities between subgroups that also appear as a mismatch between average and worst-case accuracy. In Appendix D.2 we compare post-processing methods and show the resulting trends are similar.

### 3.4.1 SKEWSIZE ACROSS VARYING MODELS

We extend the scope of our evaluation and consider all three instances of the BLIP-2 model family so that we can investigate whether models are biased in different levels, as well as whether specific characteristics such as increased scale and instruction tuning, amplify or mitigate it.

|  | Occupation | Sports |
|---|---|---|
| BLIP2-2.7B | 0.233 | 1.205 |
| BLIP2-6.7B | -0.045 | 0.360 |
| BLIP2-FlanT5 | 0.599 | 1.255 |

Table 3: **VLMs SKEWSIZE.** Measuring gender bias on occupation and sports modality prediction.

**Effect size strength.** In Figure 5, we categorize effect size values between 0 and 0.1 as negligible[3] and between 0.1 and 0.3, 0.3 and 0.5, and above 0.5 as small, medium, and large, respectively. For occupation prediction, (Fig. 6(a)), larger models have more classes which exhibit medium and large effect sizes, suggesting an overall amplification in gender bias. However, using an instruction-tuned language model leads to fewer classes with large effect in comparison to BLIP-2.7B and 6.7B, suggesting instruction tuning may mitigate bias in this instance. Results for sport modality prediction follow a similar trend (Figure 5(b)). The number of classes with negligible effect size decreases when increasing model size, while BLIP2-FlanT5 exhibits less bias.

**SKEWSIZE.** As per Section 2, a fine-grained analysis of the results in Figure 5 reveals that the empirical distribution of effect size values (Figures 6(a) and 6(b)) across all classes is heavy-tailed and left-skewed. We report in Table 3 the skewness coefficients for all models in both tasks. We find that larger models seem to exhibit more bias but instruction fine-tuning seems to mitigate the bias.

## 4 RELATED WORK

**Fairness hypothesis testing.** Previous work has proposed hypothesis testing approaches to probe for fairness under multiple definitions within both datasets (Yik et al., 2022) and algorithms (Jourdan et al., 2023). Tramer et al. (2017) introduced a permutation test based on Pearson's correlation statistic to test for statistical dependence, under a particular metric, between an algorithm's outputs and protected user groups, while DiCiccio et al. (2020) proposed a method that tests the hypothesis

---

[3]The use of the word *negligible* here *does not refer* to the extent that potential harms will affect users.

that a model is fair across two groups with respect to any given metric. Our work differs from both a methodological perspective, e.g. in comparison to Yik et al. (2022) which considers whether the data distribution is significantly different from a reference distribution, as well as applicability, since we propose a metric that can capture biases in a multi-class setting, and which goes beyond binary sensitive attributes (DiCiccio et al., 2020). Caliskan et al. (2017) proposed a permutation testing approach to evaluate bias in text based on similarity that should indicate association between words (e.g. occupation and gender) in an embedding space.

**Evaluating neural network biases.** Previous work on bias evaluation has prioritized contexts where the information necessary to measure bias can be directly inferred from text such as in the case of language generation and image search tasks (Rae et al., 2021; Wang et al., 2022a; Tang et al., 2021; Wang et al., 2021). In the case of text-to-image generative models, the corresponding attribute is usually inferred by another model (Naik & Nushi, 2023). In contrast, we evaluate bias directly in the model output space, as opposed to relying on a predicted subgroup. Birhane et al. (2023) investigated how stereotyping and bias evolves as model scale increases, finding that scale appears to amplify those issues. Luccioni et al. (2023) presented similar findings, reporting that such models over-represent white and male characteristics. Similarly, when evaluating CLIP, Radford et al. (2021); Wolfe & Caliskan (2022); Hall et al. (2023a); Prabhu et al. (2023) found that it reflected biases in the training data, exacerbated for intersectional groups. In the case of VLMs, most prior work focused on leveraging annotated datasets such as MS-COCO (Chen et al., 2015), CelebA (Liu et al., 2015) and FairFace (Karkkainen & Joo, 2021) to measure and mitigate bias (Berg et al., 2022; Chuang et al., 2023; Hall et al., 2023a), while Seth et al. (2023) and Smith et al. (2023) went further, collecting a multi-modal benchmark and using generative models to obtain contrast sets, respectively. Prior work (Zhao et al., 2017; Wang & Russakovsky, 2021) has also considered evaluating bias amplification, comparing prediction statistics with the original dataset statistics.

## 5 DISCUSSION

In this work, we propose a novel metric, SKEWSIZE, to assess biases in classification models, including when the output space is intractable. Motivated by the observation that certain biases may present in the distribution of prediction errors, we draw on tools from contingency table hypothesis testing and propose to measure bias on a per-class basis by estimating the effect size of the interaction between model prediction and the bias variable, rather than outputting a single p-value as previous work. Experiments across 4 datasets show that SKEWSIZE captures disparities that accuracy-based metrics do not surface. When the full confusion matrix is available, we also highlight that SKEWSIZE complements standard metrics like demographic parity and equalized odds by identifying classes that are potentially biased. Our results also show how SKEWSIZE can be used in practice: the per-class bias profile yielded by our approach can shed light on spurious correlations for classes presenting both higher accuracy and high effect size, while the aggregated SKEWSIZE can be used to compare models, highlighting whether increased scale amplifies biases on VLMs, and evaluate the impact of techniques as instruction tuning. Aspects to be investigated in future work include how our work could be extended to continuous attributes by using a different statistic (Brown et al., 2023). While the selected aggregation approach and metrics are suitable for the cases we consider in this work, in this case, other testing statistics or aggregations might be more appropriate.

**Limitations and Practical Considerations.** We recommend SKEWSIZE be employed alongside accuracy-based metrics for a more complete picture of a model's performance. We note that SKEWSIZE cannot infer a causal relationship between the bias attribute and model predictions, only their association. As with mathematical fairness criteria, our metric does not relate bias to potential harms (Weidinger et al., 2022); further work is needed to understand the impact of this distributional bias. The authors recognize that the binary framing of gender used in the illustrative example is an oversimplification of an important and complex topic. Our method allows for the interrogation of model bias in terms of discrete, mutually exclusive categories, which may not be ideal for representing multifaceted and intersectional human identities (see Lu et al. (2022) for an exploration of this topic). Finally, the synthetic dataset may inherit stereotypes from its generative model, e.g. misrepresenting non-cisgender people (Ungless et al., 2023). We remark that is not within the scope of our work to define which biases are practically relevant, given that this is context-dependent and that a metric should account for all existing biases in a dataset/model so that a comprehensive profile of a model's performance can be taken into consideration at the evaluation.

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

APPENDIX

## A    FAIRNESS METRICS DEFINITIONS

We consider Demographic Parity Dwork et al. (2012) as an independence fairness criterion:

$$\text{DP} = \max_{a \in \mathcal{Z}} \mathbb{E}[f_\theta(x) \mid z = A] - \min_{z \in \mathcal{Z}} \mathbb{E}[f_\theta(x) \mid z = a]. \tag{5}$$

While for separation, we refer to equalized odds (EO, Hardt et al., 2016):

$$\text{EO} = \max_{a,k \in \mathcal{Z}x\mathcal{Y}} \mathbb{E}[f_\theta(x) \mid z = a, y = k] - \min_{a,y \in \mathcal{Z}x\mathcal{Y}} \mathbb{E}[f_\theta(x) \mid z = a, y = k]. \tag{6}$$

We focus on the multi-class extension of these metrics by binarizing the task, as suggested in (Alabdulmohsin et al., 2022). The metrics are then aggregated across classes using their maximum value.

## B    COMPUTING EFFECT SIZE USING OTHER STATISTICS

In addition to the 3 accuracy based metrics and 2 fairness metrics we already considered in previous results, in this section we further include the Phi coefficient as a measure of effect size when computing SkewSize in the dSprites experiments. The results in Table 4 show that in this case the Phi Coefficient yields similar trends as the Cramer's V. Notice, however, that it is not advisable to use the Phi Coefficient on contingency tables larger than 2x2, which is the reason why we decided to use the more general Cramer's V when computing SKEWSIZEthroughout our work.

| | Cramer's V | | | Phi Coefficient | | |
|---|---|---|---|---|---|---|
| | Class 0 | Class 1 | Class 2 | Class 0 | Class 1 | Class 2 |
| Unbiased | 0.012 | 0.011 | 0.019 | 0.012 | 0.011 | 0.027 |
| Class 0 | **0.670** | 0.015 | 0.016 | **0.948** | 0.015 | 0.022 |
| Class 1 | 0.014 | **0.683** | 0.108 | 0.014 | **0.966** | 0.152 |
| Class 2 | 0.047 | 0.006 | **0.696** | 0.067 | 0.006 | **0.985** |

Table 4: Computing effect size with Cramer's V vs Phi Coefficient. dSprites dataset.

## C    EFFECT SIZE CAPTURES DIFFERENT BIAS LEVELS

We leverage the fact that our experimental setting with the dSprites dataset allows us to obtain models presenting varying levels of bias to show that our approach is capable of capturing such differences. In more detail, we consider a variation of the dSprites setting from Section 3 where we train models with subsets of different sizes containing part of the removed instances (green ellipsis/class 1 for this case) so as to introduce different bias levels. As reported in Table 5, we find that the effect size presents a monotonic relationship with bias strength, therefore a small change in the bias of the output leads to a small change in the metric and a large change in bias leads to a large change in the metric. Confirming that the effect size is not too sensitive or not sensitive enough to variations in the bias strength.

| | Accuracy metrics | | | Effect size | | |
|---|---|---|---|---|---|---|
| Bias level | Accuracy | WG | Gap | Class 0 | Class 1 | Class 2 |
| Unbiased (full training set) | 0.998 | 0.996 | 0.002 | 0.006 | 0.02 | 0.024 |
| Mildly biased (5k) | 0.977 | 0.936 | 0.041 | 0.002 | **0.253** | 0.017 |
| Highly biased (2.5k) | 0.933 | 0.806 | 0.127 | 0.013 | **0.492** | 0.032 |
| Totally biased (0) | 0.891 | 0.653 | 0.238 | 0.012 | **0.703** | 0.011 |

Table 5: Inducing different bias levels in models trained with the dSprites dataset. Bias level denotes the number of training examples from the class removed when training the totally biased model.

# D    VLM: DETAILED RESULTS

## D.1    EFFECT SIZE DISTRIBUTIONS

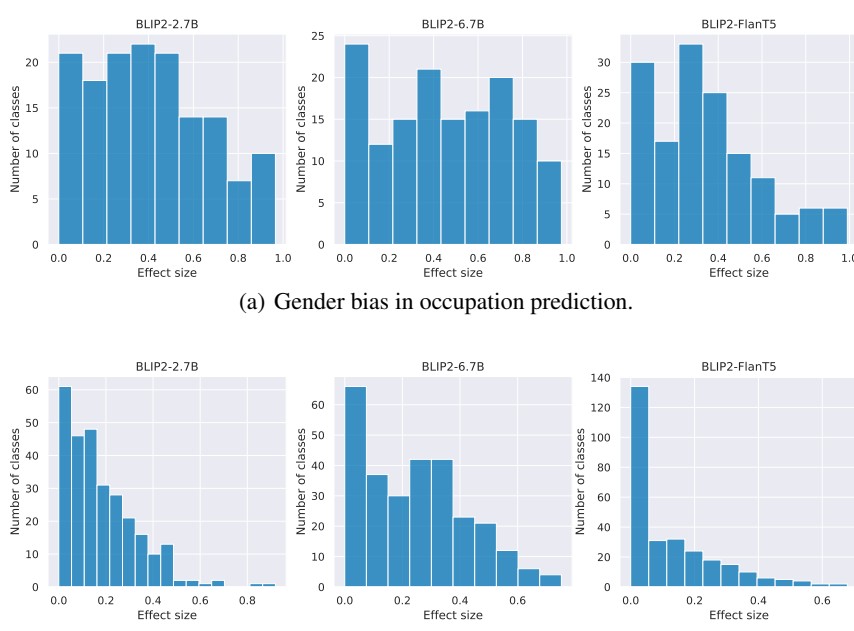

(a) Gender bias in occupation prediction.

(b) Gender bias in sport modality prediction.

Figure 6: Distribution of effect size values between gender and predicted occupation/sport modality across BLIP-2 models.

## D.2    OPTIONAL POST-PROCESSING

As we do not constrain the model's output, there may be cases where the model predicts synonyms of the ground-truth class, e.g. lawyer and attorney, or the predictions consist of sentences with different structures, e.g. *"The person is a laywer"* and *"A lawyer"*. In light of that, in order to compute accuracy values, we manually post-process the outputs of the model to account for all cases where the output semantically matched the ground-truth answer.

**Impact of post-processing.** We also investigate in Table 6 whether post-processing model outputs affects the overall experimental findings by comparing the metric trend across models for both raw and post-processed outputs. We find that the same trends can be observed irrespective of the post-processing. Increasing model size while keeping an unsupervised-trained language model amplifies bias as the skewness values decrease when comparing BLIP2-2.7B and BLIP2-6.7B (from 0.233 to -0.045). As expected, SKEWSIZE values computed with raw model outputs tend to be lower, indicating an overall increase in the computed effect size. This is because, without post-processing, the predicted classes are more fine-grained, resulting in a potential larger mismatch between predictions for each gender value. BLIP2-FlanT5 presented the highest skewness values for all cases, further confirming the findings in Figure 5.

|  | Raw | Occupation |
|---|---|---|
| BLIP2-2.7B | ✗ | 0.233 |
|  | ✓ | -0.005 |
| BLIP2-6.7B | ✗ | -0.045 |
|  | ✓ | -0.130 |
| BLIP2-FlanT5 | ✗ | 0.599 |
|  | ✓ | 0.124 |

Table 6: **SKEWSIZEfor raw versus post-processed model outputs**. Higher skewness values correspond to models having less gender bias. We observe that post-processing the models outputs changes the skewness value but *does not* change the overall trend.

### D.3 CONTROLLING THE EFFECT OF NOISE IN THE PREDICTIONS

As the output space grows, it is possible that the metric would become more sensitive. In order to limit this effect, we have used the following strategy: as per the rule-of-thumb to satisfy the assumption of the Chi-square test, we can remove columns from the contingency with respective expected value lower than 5. As we are looking for systematic patterns in the errors of the model, using such a filtering strategy reduces sensitivity to randomness while maintaining sensitivity to the systematic patterns. We can also vary this value in order to decide to which degree some randomness in the predictions should be taken into account

To illustrate how the choice of the minimum expected value to be accounted for would affect results, we repeated the evaluation reported in Section 3.4.1 for the occupation prediction task with varying thresholds so that we can evaluate whether the comparison between models would change. As demonstrated by the results in Table 7, the choice of threshold does not affect the resulting comparison between models.

|  | MEV=6 | MEV=5 | MEV=4 | MEV=3 | MEV=2 |
|---|---|---|---|---|---|
| BLIP-2.7 | 0.235 | 0.233 | 0.225 | 0.199 | 0.19 |
| BLIP-6.7 | -0.031 | -0.045 | -0.056 | -0.072 | -0.102 |
| BLIP-FlanT5 | 0.625 | 0.599 | 0.578 | 0.544 | 0.507 |

Table 7: Varying the minimum expected value (MEV) for evaluating the BLIP model family in the occupation prediction task.

### D.4 DATA GENERATION

We consider 148 and 273 classes for the tasks of occupation and sport modality prediction, respectively. The complete list of occupations and sport modalities used in the VLMs experiments can be found in the Supplementary Material repository.

## E IMAGENET MODELS

We used a variety of models trained on IMAGENET with different sizes, training accuracy, pretraining, etc. Unless otherwise stated, we used publicly available models from TF-HUB[4].

- RESNET50-1/2 (He et al., 2016): A model we trained on IMAGENET from scratch which achieved around 76% accuracy.
- RESNET* (He et al., 2016): RESNET models with no pretraining.
- VIT* (Dosovitskiy et al., 2020): A B/16 variant of the vision transformer model family we trained on IMAGENET from scratch which achieved around 80% accuracy.
- INCEPTION* (Szegedy et al., 2015): Inception models with no pretraining.
- INCEPTION RESNET (Szegedy et al., 2017): A hybrid INCEPTION RESNET model with no pretraining.
- BIT-S* (Kolesnikov et al., 2020): BIT models with no pretraining.

---

[4]https://tfhub.dev/google/imagenet/

## F  FURTHER RELATED WORK

**Mitigations.**   Given a known bias in the model, it is possible to mitigate the issue, demonstrating the importance of being able to identify biases to improve the model. This can be done by intervening on the dataset to make it fairer while maintaining performance as done by Singla et al. (2022). Another approach is to intervene on the prompts and de-bias the text embeddings as done by Chuang et al. (2023). Finally, we can intervene at the model level, as done by Friedrich et al. (2023); Berg et al. (2022) and use guidance or an adversarial loss to steer the model towards being more fair. (Zhang et al., 2018), (Alvi et al., 2018) (Kim et al., 2023), (Li et al., 2022)

**Datasets for evaluating bias.**   To develop methods to evaluate and mitigate bias, datasets such as Waterbirds (Sagawa et al., 2019), CelebA (Liu et al., 2015), and MultiNLI (Williams et al., 2017) have been used; in these datasets the biases are created a-priori (e.g. land birds on land backgrounds vs water birds on watery backgrounds). Revise was introduced by Wang et al. (2022b) in order to visualise the biases in a dataset (and thereby probable impacts on models trained on such a dataset). However, such a tool requires labels on what exists in the dataset, which may not be possible, so Jain et al. (2022) demonstrated how biases could be found automatically in multimodal datasets. A targeted dataset looking at model performance at predicting everyday objects conditioned on geographical location is the DollarStreet dataset (Rojas et al., 2022). Other targeted datasets, such as FairFaces and CasualConversationsV2 (Karkkainen & Joo, 2021; Porgali et al., 2023; Hall et al., 2023b) can be used to evaluate a models bias across sensitive attributes. Karkkainen & Joo (2021); Porgali et al. (2023) does so by comparing classification performance across these attributes (e.g. age, gender, etc.) whereas Hall et al. (2023b) is a small dataset of 250 images that evaluates pronoun resolution conditioned on the image. Such datasets are complementary to our approach, as we rely on a dataset with labelled subgroups to compute our metric. When no such dataset exists, we relied on synthetic data.

## G  SKEWSIZE IMPLEMENTATION DETAILS

PSEUDOCODE

---
**Algorithm 1** Computing SKEWSIZE
---
1: **for** $i = 1, 2, \ldots, |\mathcal{Y}|$ **do**
2:     Get set of model predictions $\hat{Y}^i = \{\hat{y}_k\}$ for all $(x_k, y_k, z_k)$ where $y_k = y_i$
3:     **for** $j = 1, 2, \ldots, |\mathcal{Z}|$ **do**
4:         Build $\hat{Y}^{ij}$, a subset of $\hat{Y}^i$ with instances where $z_k = z_j$
5:     **end for**
6:     Estimate $\nu_i$, the effect size for the $i$-th class, using Equation 2
7: **end for**
8: Aggregate effect size estimates per class by computing SKEWSIZE as per Equation 4
---

PYTHON IMPLEMENTATION

```python
# Copyright 2023 The SkewSize Authors. All rights reserved.
# SPDX-License-Identifier: Apache-2.0

import numpy as np
import pandas as pd
import scipy.stats as stats

v_list = []
for label in unique_labels:
    # predictions: predictions for all instances in the class *label*.
    # subgroups: predictions for all instances in the class *label*.
    df = pd.DataFrame({'predictions': predictions,
                       'subgroups': subgroups})
    crosstab = pd.crosstab(df.subgroups, df.predictions)

    chi2 = stats.chi2_contingency(crosstab)[0]
    dof = min(crosstab.shape)-1
    n = crosstab.sum().sum()
    v = np.sqrt(chi2/(n*dof))
    v_list.append(v)

v_values = np.asarray(v_list)
# When a model predicts correctly all examples
# in a given class across all subgroups
# dof=0 and the corresponding v is NaN.
# We remove NaNs before computing skewsize.
v_values = v_values[~np.isnan(v_values)]
skewsize = stats.skew(v_values)
```

