# OpenReview forum: "Evaluating model bias requires characterizing model mistakes"
_ICLR.cc/2024/Conference — Submitted to ICLR 2024_

### Official Review · Reviewer_bd8T · 2023-10-14

**Soundness:** 3 good
**Presentation:** 2 fair
**Contribution:** 3 good
**Rating:** 8
**Confidence:** 2

**Summary:**

This paper is about the evaluation of machine learning models to detect undesirable behaviour, such as inconsistent behaviour across subgroups of the test data. The contribution is a novel metric named SkewSize. It is a scalar measuring the skewness across classes of the effect size of subgroup identity (based on some biased/sensitive attribute) on the model's predictions. The metric is shown to be applicable in a variety of settings, and its utility is demonstrated with recent existing vision and vision-and-language models.

**Strengths:**

- Clear motivation, important problem. The need for finer-grained evaluation of models is clearly shown with the motivating example that exposes how existing metrics can be misleading.

- The novel metric seems sounds and grounded in solid statistical principles.

- The applicability to various settings is demonstrated both in a controlled environment (dSprites) and in a practical/realistic scenario (vision and V&L models).

**Weaknesses:**

I see no clear flaws in this paper.

**Questions:**

N/A. This seems to be a strong paper: it addresses an important problem that seems to have been overlooked, with a method that appears technically sound, and that is demonstrated in both controlled and realistic settings.

Minor suggestions:
- "*quantify the numbers*" -> "aggregate"
- "open-ended predictions": need a dash
- "visual language models", "vision language models", ... I think the most common term is "vision-and-language models"
- Sect 2.1: "On the other hand": should only be used if there was another statement before starting with "on the one hand". I think that here you can replace it with "Alternatively".

---

> ### Author Response · Authors · 2023-11-16
> **Rebuttal**
>
> Thank you for taking the time to review our paper. We appreciate your constructive feedback and the positive comments on our paper and proposed metric! We have addressed the suggestions and did another thorough proof-reading of the paper.

---

### Official Review · Reviewer_kSV6 · 2023-10-23

**Soundness:** 3 good
**Presentation:** 4 excellent
**Contribution:** 2 fair
**Rating:** 6
**Confidence:** 4

**Summary:**

The paper introduces a novel metric, SKEWSIZE, designed to identify biases in a model's predictions. The authors first establish the need for such a metric by providing several examples where existing metrics fall short in capturing inherent biases. The proposed SKEWSIZE metric is applicable in a variety of scenarios, including synthetic data, distribution shift/multi-class classification, and open-ended predictions. This wide applicability is demonstrated through a series of experiments in Section 3.

**Strengths:**

1. The paper is well-structured and comprehensive, beginning with various motivational examples and addressing the shortcomings of existing metrics. The authors then propose their method and try to demonstrate that SKEWSIZE outperforms previous metrics in more practical, large-scale scenarios via experiments.

2. The research question - seeking a better metric to capture model bias - is clearly defined and of importance in the field.

3. The experiments conducted demonstrate the versatility of the proposed metric, showing its applicability across different tasks.

**Weaknesses:**

1. The novelty of the proposed metric is a concern. The use of contingency tables, relevant statistics, and skewness for data analysis is not a new thing. It appears that SKEWSIZE merely combines these elements to measure fairness/bias. Could the authors elucidate further on what makes their proposed metric unique?

2. The discussion in Section 2.2 is constrained to cases with only two items in Z. Is it feasible to extend this discussion to scenarios with multiple items?

3. Regarding the experiment section, it's unclear how computing the correlation between SKEWSIZE and other metrics, as done in Sections 3.2 and 3.3, demonstrates "that the proposed metric identifies issues with models not exposed by these other metrics".

**Questions:**

See weaknesses.

**Details Of Ethics Concerns:**

Although the paper is related to "societal considerations including fairness, safety, privacy", it doesn't seem to raise any ethical concerns based on the content presented.

---

> ### Author Response · Authors · 2023-11-16
> **Rebuttal part 1**
>
> Thank you for taking the time to review our paper. We appreciate your constructive feedback and we address below the points raised in your review:
>
> *1. “The novelty of the proposed metric is a concern. The use of contingency tables, relevant statistics, and skewness for data analysis is not a new thing. It appears that SKEWSIZE merely combines these elements to measure fairness/bias. Could the authors elucidate further on what makes their proposed metric unique?”*
>
> Our main contribution lies in proposing a principled and not yet seen in the literature approach to leverage established statistical methods to characterize biases in neural networks across multiple settings, including cases where models are used in open-ended ways such as in Section 3.4 of our work. We agree with the reviewer that the separate statistical elements are not novel.  As we describe in the Related Works section, prior work has also investigated statistical methods to investigate fairness, but these techniques had limitations such as providing  a single p-value [1]. In contrast, we show that by referring to the effect size, and using the skewness to aggregate results across classes, we obtain both a detailed and a global view of the bias. In addition, we believe that referring to well-established statistical methods is a strength of this work, as it is principled and provides guidance on which method should be used to measure effect size and how.
>
> We have added these elements in the abstract and discussion:
> Highlighting in the abstract that our main contributions are related to propose a principled approach to measure bias that leverages well-established statistical analysis techniques: “Inspired by the well-established hypothesis testing framework, we introduce \metricname, a principled and flexible metric that captures bias from mistakes in a model's predictions.”
> Further highlighting in the discussion (Section 5) that our metric adds to the literature of statistical methods to measure fairness by considering the effect size between interactions rather than p-values. By doing so we can provide both a per-class and global view of a model’s bias profile: “we draw on tools from contingency table hypoth-esis testing and propose to measure bias on a per-class basis by estimating the effect size of the interaction between model prediction and the bias variable, **rather than outputting a single p-value as previous work**”.
>
> [1] Tramer et al., “Fairtest:  Discovering  unwarranted  associations  in  data-driven applications”, 2017.
>
> *2-“The discussion in Section 2.2 is constrained to cases with only two items in Z. Is it feasible to extend this discussion to scenarios with multiple items?”*
>
> Thank you for this comment. We decided to use binary attributes in Definition 2.2 just for the sake of clarity in the presentation. While the definition of distributional bias employs the binary case, the $\mathcal{H}$ operator is not limited to binary attributes as we can instantiate it by using the Cramer’s V effect size, which works for contingency tables of any size.
>
> Following the reviewer’s comment, we have added the following text to Section 2.2 in order to clarify this point:
>  “Notice that the $\mathcal{H}$ operator is not limited to binary attributes and can be instantiated by approaches to simultaneously compare multiple families of distributions.”

---

> ### Author Response · Authors · 2023-11-16
> **Rebuttal part 2**
>
> *3-“Regarding the experiment section, it's unclear how computing the correlation between SKEWSIZE and other metrics, as done in Sections 3.2 and 3.3, demonstrates "that the proposed metric identifies issues with models not exposed by these other metrics".”*
>
> We compute the correlation between different metrics to highlight how these are related to each other. For instance, Section 3.1 displays that EO will capture bias, but not necessarily on the class it originates from. Therefore, the correlation per class between these two metrics shows that they both can highlight biases, but they are not equivalent. When compared to other metrics such as GAP or worst group accuracy, the correlation is not significant, demonstrating that these metrics measure  fundamentally different aspects of bias. In particular, GAP or worst group accuracy are not sufficient to evaluate bias arising from the mispredictions of a model as demonstrated in Figure 1 in the paper. The fact that in Sections 3.2 and 3.3 this leads to different models having lower SkewSize than Accuracy (as demonstrated by the correlation), again highlights that SkewSize is exposing biases that are not captured by the accuracy measurements. Therefore, we claim that Effect size and Skewsize “provides complementary information to other metrics” (as mentioned in the results and in the discussion).
>
> We have added the following sentence to Section 3.2 in order to make this point clearer and address the reviewer’s concern: “[the correlation results...] indicate the metrics are related but not equivalent as they capture distinct aspects of the bias.”
>
> We would greatly appreciate it if you let us know in case there are any other questions so we can further improve the manuscript based on your feedback.

---

> > ### Comment · Reviewer_kSV6 · 2023-11-17
> > **Response to Rebuttal**
> >
> > Thanks for the authors' response. However, I still hold some reservations about the third point.
> >
> > I understand that "the correlation results indicate the metrics are related but not equivalent". But I cannot understand why the insignificant correlation can tell us "these metrics measure fundamentally different aspects of bias". Could the authors provide more explanations for this? Can we have some examples to understand what "fundamentally different aspects of bias" different measurements are revealing respectively?
> >
> > Authors said "GAP or worst group accuracy are not sufficient". Given the lack of a fairness benchmark, it's challenging, in my view, to determine the sufficiency of any measure. It would be more persuasive to present experimental results in controlled environments where we have some prior knowledge of fairness. The experiment in Section 3.1 is a good example. Could larger-scale controlled experiments be implemented for example using Stable Diffusion to generate controlled image samples and LLMs to create controlled language samples? I'm curious to hear the authors' opinions on this.

---

> > > ### Author Response · Authors · 2023-11-17
> > > **Response to new comments**
> > >
> > > Thank you for the quick response and for engaging in the discussion! We are glad the first two points have been addressed by our previous responses and appreciate your feedback. Below you can find our responses to the most recent comments:
> > >
> > > *"I understand that "the correlation results indicate the metrics are related but not equivalent". But I cannot understand why the insignificant correlation can tell us "these metrics measure fundamentally different aspects of bias". Could the authors provide more explanations for this? Can we have some examples to understand what "fundamentally different aspects of bias" different measurements are revealing respectively?"*
> > >
> > > We apologize for the confusion. What we mean is that, starting from the hypothesis that the metrics capture different aspects of the model output, i.e. are not equivalent (which is given by their respective definitions), the observed mild correlation between them indicates the hypothesis is correct. On the other hand, in case all metrics were capturing the same performance aspects they would be highly correlated. We hope this clarifies the point we made in the text. In order to mitigate this confusion in the manuscript we modified the text in the Section 3.2 from “which indicate the metrics are related but not equivalent as they capture distinct aspects of the bias”, to:
> > >
> > > “which indicate the metrics are related but not equivalent as in case they were capturing the same aspects of the model performance they should be highly correlated”.
> > >
> > > Please refer to the next answer in order to see examples where effect size reveals different aspects of bias in comparison to other metrics.
> > >
> > > *"Authors said "GAP or worst group accuracy are not sufficient". Given the lack of a fairness benchmark, it's challenging, in my view, to determine the sufficiency of any measure. [...] The experiment in Section 3.1 is a good example. Could larger-scale controlled experiments be implemented for example using Stable Diffusion to generate controlled image samples and LLMs to create controlled language samples? I'm curious to hear the authors' opinions on this.”*
> > >
> > > Thank you for raising this interesting point. As in settings with real-world datasets and larger models, bias might not only be a function of how unbalanced the training data is [1], we believe coming up with a larger-scale controlled experiment akin to what we have in Section 3.1 would be far from trivial. But in order to address the reviewer’s concerns regarding cases where GAP or worst group accuracy are not sufficient (and also to highlight the differences between metrics), we revisit our experiments with large-scale VLMs and perform a fine-grained evaluation of some of the classes reported in Table 2.
> > >
> > > The following three tables show the number of predictions by BLIP-FlanT5 for the top-4 predicted classes when evaluating images corresponding to male and female people in the classes Writer, Veterinarian, and Teachers, respectively. As in the experiments reported in the paper, 500 images per subgroup and occupation were evaluated.
> > >
> > > We observe that for all three cases the model distributes the mistakes across different classes per subgroup. For example, in the case of the *Writer* class, the similar number of correct predictions for both male and female subgroups makes accuracy considerably high and the accuracy gap low, which could make one mistakenly conclude this model does not exhibit bias in this setting. However, by observing **the distribution of incorrect predictions**, it is easy to see the model demonstrates to be biased towards **mistaking female writers by teachers or singers, whereas male writers are more often confused by composers**. While both **accuracy-based metrics do not capture this behavior**, the **effect size does**. This can be seen by the respective value equal to 0.269, which indicates a non-negligible interaction between gender and predictions for this class.
> > >
> > > **Writer:**
> > > Accuracy: 80.24%
> > > Gap: 0.6%
> > > Effect size: 0.269
> > >
> > > |        | Writer | Teacher | Singer | Composer |
> > > |--------|--------|---------|--------|----------|
> > > | Male   | **395**    | 8       | 0      | 31       |
> > > | Female | **401**    | 27      | 25     | 1        |
> > >
> > > **Veterinarian:**
> > > Accuracy: 79.13%
> > > Gap: 0.10%
> > > Effect size: 0.154
> > >
> > > |        | Veterinarian | Doctor | Nurse | Pharmacist |
> > > |--------|--------------|--------|-------|------------|
> > > | Male   | **390**          | 96     | 5     | 2          |
> > > | Female | **393**          | 66     | 25    | 9          |
> > >
> > >
> > > **Teacher**
> > > Accuracy: 74.40%
> > > Gap: 8.47%
> > > Effect size: 0.225
> > >
> > > |        | Teacher | Lawyer | Businessperson | Writer |
> > > |--------|---------|--------|----------------|--------|
> > > | Male   | **327**     | 39     | 41             | 13     |
> > > | Female | **411**     | 8      | 14             | 18     |
> > >
> > > [1] Hooker, Sara. "Moving beyond “algorithmic bias is a data problem”." Patterns 2.4 (2021).

---

> > > > ### Comment · Reviewer_kSV6 · 2023-11-19
> > > >
> > > > Thanks for the authors' further response.
> > > >
> > > > I feel the examples provided well illustrate how better *Effect size* is to reveal model bias than other metrics. I would recommend including these examples in the revised version of the paper.
> > > >
> > > > I raised my rating to 6.

---

### Official Review · Reviewer_1DWK · 2023-10-29

**Soundness:** 3 good
**Presentation:** 4 excellent
**Contribution:** 2 fair
**Rating:** 5
**Confidence:** 4

**Summary:**

This paper introduces the SkewSize metric, an approach for characterizing model bias by accounting for systematic mismatches in prediction errors across subgroups. The metric provides a more comprehensive evaluation of model bias than traditional metrics, such as worst-group accuracy or accuracy gap, and is demonstrated to be effective in various settings, including vision models and vision-language models, uncovering biases that other metrics might miss.

**Strengths:**

- SkewSize is demonstrated to be applicable in a variety of settings, including vision models, domain shift benchmarks, and large-scale vision-language models. Its versatility and effectiveness make it a valuable tool for assessing biases in different contexts.
- Model bias is a crucial concern in practical applications of machine learning. The paper's focus on characterizing and quantifying bias is highly relevant in building fair and trustworthy AI systems.

**Weaknesses:**

- The paper's SkewSize metric is based on statistical concepts such as skewness and chi-square tests, which may not represent a fundamentally novel approach. While it introduces a new metric, the underlying statistical principles are not groundbreaking.
- The paper may lack excitement or the potential to generate significant interest, as it builds upon established statistical techniques rather than introducing innovative or revolutionary methodologies.
- While the paper identifies biases in model predictions using the SkewSize metric, it doesn't definitively confirm whether these biases are genuine or artifacts. The sensitivity of the metric could potentially lead to the detection of biases that are not practically significant.
- There's a concern that the SkewSize metric might be overly sensitive to minor variations or noise in data, which could result in an overemphasis on biases that may not be robust or practically impactful.
- The paper doesn't thoroughly address the potential for false positives when using the SkewSize metric. It's crucial to differentiate between genuine biases and statistical anomalies.
- The paper doesn't provide a strong discussion of the ethical or practical implications of its findings. While it highlights biases, it may not offer clear guidance on how to address or mitigate these biases in real-world applications.

**Questions:**

- In the code implementation of SkewSize, why the degree of freedom is calculated by the minimum of the prediction class and subgroup class minus 1? Shouldn't it be $(prediction class - 1)*(subgroup class - 1)$?
- When there exists a large group of object classes, the effect size may be large by chance, hence the calculated SkewSize will tend to be skew. I wonder if the authors have conducted experiments to understand the robustness of the proposed metric. SkewSize seems to be very sensitive in that it can identify potential bias in the model predictions. However, it is not clear that in the case where no bias exists in the predictions, SkewSize also demonstrates some degree of skewness.

---

> ### Author Response · Authors · 2023-11-16
> **Rebuttal part 1**
>
> Thank you for taking the time to review our paper. We appreciate your constructive feedback and we address below the points raised in your review:
>
> **Weaknesses**
>
> *“While it introduces a new metric, the underlying statistical principles are not groundbreaking. The paper may lack excitement or the potential to generate significant interest, as it builds upon established statistical techniques rather than introducing innovative or revolutionary methodologies.”*
>
> Our main contribution lies in proposing a principled approach not yet seen in the literature : we leverage established statistical methods to characterize biases in neural networks across multiple settings, including cases where models are used in open-ended ways (such as in Section 3.4 of our work). We agree with the reviewer that the separate statistical elements are not novel.  As we describe in the Related Works section, prior work has also investigated statistical methods to investigate fairness, but these techniques had limitations such as providing  a single p-value [1]. In contrast, we show that by referring to the effect size, and using the skewness to aggregate results across classes, we obtain both a detailed and a global view of the bias. In addition, we believe that referring to well-established statistical methods is a strength of this work, as it is principled and provides guidance on which method should be used to measure effect size and how.
>
> We have added these elements in the abstract and discussion:
>
> - Highlighting in the abstract that our main contributions are related to propose a principled approach to measure bias that leverages well-established statistical analysis techniques: “Inspired by the well-established hypothesis testing framework, we introduce SkewSize, a principled and flexible metric that captures bias from mistakes in a model's predictions.”
> - Further highlighting in the discussion (Section 5) that our metric adds to the literature of statistical methods to measure fairness by considering the effect size between interactions rather than p-values. By doing so we can provide both a per-class and global view of a model’s bias profile: “we draw on tools from contingency table hypothesis testing and propose to measure bias on a per-class basis by estimating the effect size of the interaction between model prediction and the bias variable, **rather than outputting a single p-value as previous work**”.
>
> [1] Tramer et al., “Fairtest:  Discovering  unwarranted  associations  in  data-driven applications”, 2017.

---

> ### Author Response · Authors · 2023-11-16
> **Rebuttal part 2**
>
> **Weaknesses (cont.)**
>
> *“While the paper identifies biases in model predictions using the SkewSize metric, it doesn't definitively confirm whether these biases are genuine or artifacts. The sensitivity of the metric could potentially lead to the detection of biases that are not practically significant.”*
>
> We believe there is some misunderstanding with respect to how the reviewer is interpreting SkewSize. The metric **is designed** to be sensitive to different degrees of bias and this should be reflected in the strength of the effect size for each evaluated class. In more detail, we propose a metric that accounts for the effect size of the interaction between the bias and prediction variables, i.e. how much bias there is in the predictions across classes, and we **do not set a threshold** so as to deem classes/models as positive/negative biased.
>
> To further elucidate this point, we leverage the fact that our experimental setting with the dSprites dataset allows us to obtain models presenting varying levels of bias to show that our approach is capable of capturing such differences. In more detail, we consider a variation of the dSprites setting from Section 3.1 of our manuscript where we train models with subsets of different sizes containing part of the removed instances (green ellipsis/class 1 for this case) so as to introduce different bias levels. As reported in the table below (Table 5 in the revised version of the manuscript), we find that the effect size presents a monotonic relationship with bias strength, therefore a small change in the bias of the output leads to a small change in the metric and a large change in bias leads to a large change in the metric. **Confirming that the effect size is not too sensitive or not sensitive enough to variations in the bias strength.**
>
> |                              | Accuracy metrics |       |       |  Effect size |         |         |
> |:----------------------------:|:----------------:|:-----:|:-----:|:-----------:|:-------:|:-------:|
> |          Bias level          |     Accuracy     |   WG  |  Gap  |   Class 0   | Class 1 | Class 2 |
> | Unbiased (full training set) |       0.998      | 0.996 | 0.002 |    0.006    |   0.02  |  0.024  |
> |      Mildly biased (5k)      |       0.977      | 0.936 | 0.041 |    0.002    |  0.253  |  0.017  |
> |     Highly biased (2.5k)     |       0.933      | 0.806 | 0.127 |    0.013    |  0.492  |  0.032  |
> |      Totally biased (0)      |       0.891      | 0.653 | 0.238 |    0.012    |  0.703  |  0.011  |
>
> Finally, we remark that,  in our opinion, the definition of practically relevant bias is context-dependent and that a metric should account for all existing biases in a dataset/model so that a comprehensive profile of a model's performance can be taken into consideration at the evaluation. In case, for example, biases associated with negligible effect size are not practically relevant for a particular application, the corresponding classes can be excluded from the aggregation step when computing SkewSize.
>
> To reflect in the manuscript our response to the pertinent point raised by the reviewer we have added the table along with the respective discussion to the Appendix C.
>
> *“While it highlights biases, it may not offer clear guidance on how to address or mitigate these biases in real-world applications.”*
>
> Our work adds to the bias mitigation literature in that it proposes a statistically grounded metric that captures distributional bias as defined in our manuscript. In terms of a bias mitigation pipeline, evaluating and comparing models with SkewSize can be seen as the initial step to be taken when considering attenuating biases in the outputs of a model. SkewSize can already be used out of the box for model selection. We note this in the VLM part of the paper (c.f. Section 3.4.1) that using an instruction tuned model seems to lead to less bias than the base model. Similarly, in the ImageNet case, we can choose a model with similar accuracy but less bias (give example). Another step would be to use the metric for hill climbing, but we note that how to balance trade-offs between this metric and another accuracy based metric is beyond the scope of this work.
>
> An additional benefit of our approach is that typical metrics provide only one value across all classes [2], while the effect size allows one to investigate which classes are more affected by the bias. Hence Skewsize provides additional information and insights to target the mitigation, When the full confusion matrix is available, we also highlight that SkewSize complements standard fairness metrics like demographic parity and equalized odds by identifying classes that are potentially biased, which can guide mitigation strategies by focusing on these classes.
>
> [2] Ibrahim Alabdulmohsin, Jessica Schrouff, and Oluwasanmi O Koyejo. “A reduction to binary ap-proach for debiasing multiclass datasets.” NeurIPS, 2022.

---

> ### Author Response · Authors · 2023-11-16
> **Rebuttal part 3**
>
> **Weaknesses (cont.)**
>
> *“The paper doesn't provide a strong discussion of the ethical or practical implications of its findings.”*
>
> We thank the reviewer for the comment and remark that we discuss ethical or practical implications of our work throughout the text and in the Limitations section. We note, for instance, that we limit our scope to devising and validating a metric to expose bias and do not speak to which biases should be removed or which can be tolerated (see footnote 3 on page 8). We further discuss the implications of limitations such as the use of a binary attribute to denote gender, which is not ideal for representing multifaceted and intersectional human identities.
>
> In order to further enrich the discussion of the ethical and practical implications of our findings, we added the following text to the subsection (renamed so as to reflect the reviewer’s concerns) “Limitations and Practical Considerations” in Section 5:
>
> “We remark that is not within the scope of our work to define which biases are practically relevant, given that this is context-dependent and that a metric should account for all existing biases in a dataset/model so that a comprehensive profile of a model's performance can be taken into consideration at the evaluation.”
>
> **Questions**
>
> *“In the code implementation of SkewSize, why the degree of freedom is calculated by the minimum of the prediction class and subgroup class minus 1? Shouldn't it be $(predictionclass-1)*(subgroupclass-1)$?”*
>
> We compute the empirical Cramer’s V statistics as per the definitions in [3,4,5].
>
> [3] Cramer, Harold. “Mathematical methods of statistics”. Princeton Press, NJ, 1946 \
> [4] Bergsma, Wicher. "A bias-correction for Cramér’s V and Tschuprow’s T." Journal of the Korean Statistical Society, 2013. \
> [5] [Wikipedia page for Cramer's V](https://en.wikipedia.org/wiki/Cram%C3%A9r%27s_V#:~:text=Cram%C3%A9r%27s%20V%20varies%20from%200,of%20their%20maximum%20possible%20variationRe).
>
> *“When there exists a large group of object classes, the effect size may be large by chance, hence the calculated SkewSize will tend to be skew. I wonder if the authors have conducted experiments to understand the robustness of the proposed metric.”*
>
> We thank the reviewer for this important question. We agree that, as the output space grows, it is possible that the metric would become more sensitive. In order to limit this effect, we have used the following strategy: as per the rule-of-thumb to satisfy the assumption of the Chi-square test, we can remove columns from the contingency with respective expected value lower than 5.  As we are looking for ‘systematic’ patterns in the errors of the model, using such a filtering strategy reduces sensitivity to randomness while maintaining sensitivity to the systematic patterns. We can also vary this value in order to decide to which degree some randomness in the predictions should be taken into account. We do agree with the reviewer that this is an important point and we have decided to expand on this in the main text in Section 2.2.
>
> To illustrate how the choice of the minimum expected value to be accounted for would affect results, we repeated the evaluation reported in Section 3.4.1 for the occupation prediction task with varying thresholds so that we can evaluate whether the comparison between models would change. As demonstrated by the results in Table 3, the choice of threshold does not affect the resulting comparison between models.
>
> |             |  MEV=6 |  MEV=5 |  MEV=4 |  MEV=3 |  MEV=2 |
> |:-----------:|:------:|:------:|:------:|:------:|:------:|
> |   BLIP-2.7  |  0.235 |  0.233 |  0.225 |  0.199 |  0.19  |
> |   BLIP-6.7  | -0.031 | -0.045 | -0.056 | -0.072 | -0.102 |
> | BLIP-FlanT5 |  0.625 |  0.599 |  0.578 |  0.544 |  0.507 |
>
> We have added the table and corresponding discussion to the Appendix D.3 and also have clarified this aspect in Section 2.3 by adding the following: “In the Appendix we provide pseudocode for SkewSize (Algorithm 1), a Python implementation, **and a discussion on modulating the impact low count predictions have when computing SkewSize**”

---

> ### Author Response · Authors · 2023-11-16
> **Rebuttal part 4**
>
> **Questions (cont.)**
>
> *“I wonder if the authors have conducted experiments to understand the robustness of the proposed metric. SkewSize seems to be very sensitive in that it can identify potential bias in the model predictions. However, it is not clear that in the case where no bias exists in the predictions, SkewSize also demonstrates some degree of skewness.”*
>
> We hope that, based on the expanded Section 3.1 and the responses above, the reviewer will agree that SkewSize does not demonstrate high sensitivity and is in line with expectations when results are unbiased. In fact, in the initial version of the our manuscript we have investigated this aspect in Section 3.1 as the experimental setup allowed us to control for bias in the sense that we trained the model with a perfectly balanced dataset and then compute the effect size values for all classes (c.f. results in Table 1 in the manuscript). **Results show that when there is no bias, the effect size is small for all classes.**
>
> In addition to the experiment discussed above, the result reported in part 2 of the rebuttal further corroborates our point by showing that the effect is able to capture different levels of bias induced in the model.
>
> We would greatly appreciate it if you let us know in case there are any other questions so we can further improve the manuscript based on your feedback.

---

> > ### Author Response · Authors · 2023-11-20
> > **Gentle nudge**
> >
> > Dear Reviewer 1DWK,
> >
> > Thank you for your work in reviewing our paper. We have responded to your review, performed new experiments in light of your comments, and revised our paper to reflect your feedback. As the discussion period is approaching its end, we would appreciate it if you could go through our response, consider revising your review/score accordingly or let us know in case you have further questions.
> >
> > Kind regards,
> > Authors

---

> > > ### Comment · Reviewer_1DWK · 2023-11-23
> > >
> > > Thank you to the authors for their response. My primary concern centers around the novelty of the proposed metrics, primarily relying on traditional statistical concepts such as skewness and chi-square tests. In prediction tasks involving thousands of objects, like those in ImageNet, the effect size may appear significant by chance, leading to skewed SkewSize calculations. While the paper introduces the SkewSize metric, it could potentially be replaced by just a chi-square test, offering a p-value to convey the significance of skewness. The paper does not thoroughly address the risk of false positives associated with the SkewSize metric. It is imperative to distinguish between genuine biases and statistical anomalies for a more comprehensive evaluation.

---

> ### Author Response · Authors · 2023-11-23
>
> We thank the reviewer for engaging in the discussion. Please see our detailed responses below. We hope this clarifies our contributions, and addresses the concerns related to sensitivity. We welcome further comments or feedback, and would greatly appreciate a reconsideration of your score if you are satisfied with our responses.
>
> **My primary concern centers around the novelty of the proposed metrics, primarily relying on traditional statistical concepts such as skewness and chi-square tests.**
>
> As mentioned in our prior response, we believe this is a strength of the method, and opens the avenue for more usage of statistical methods to assess distributional biases. We would like to emphasize that, through the use of classical statistical methods, our method is able to surface distributional biases that *no other fairness or robustness method can highlight.* We further illustrate how these biases arise in large vision language models. Therefore, we believe our contribution is important for the development of fairness and robustness metrics for models with open-ended vocabularies.
>
>
> **In prediction tasks involving thousands of objects, like those in ImageNet, the effect size may appear significant by chance, leading to skewed SkewSize calculations.[...] The paper does not thoroughly address the risk of false positives associated with the SkewSize metric. It is imperative to distinguish between genuine biases and statistical anomalies for a more comprehensive evaluation.**
>
> A key aspect of our method to deal with this issue is the use of the minimum expected value (MEV). We display in our prior response the impact of this hyper-parameter on our analyses. To provide a more thorough evaluation of false positives, we performed additional simulations by generating random predictions for 2 groups. Each group is drawn from the same distribution, and hence a small effect size is expected. We generate 2,000 samples per group, and vary the number of classes from 2 to 800 with a step of 20. For each sampling and number of classes, we compute the effect size, for MEV = 0 and for MEV = 5, repeating the operation 20 times.
>
> We then display the mean and std across trials of the effect size across the number of classes. We observe that the MEV allows to limit the effect size and reaches a plateau across the number of classes. On the other hand, p-values display more stability across number of classes, with non-significant results. While p-values did not reflect the strength of the bias in our prior experiments when bias was indeed present (not shown), we will suggest to consider the effect size for significant tests only. This should prevent false positives and reduce the sensitivity of our metric. We will add this analysis to our methods (section 2.2), and the Appendix.
>
> [Link to Figure: effect size vs number of classes](https://drive.google.com/file/d/15pZeBhrwq9SmkVrh4tEoMUdmkp4PjFAj/view?usp=sharing)
>
> [Link to Figure: p-value vs number of classes](https://drive.google.com/file/d/1-pXxgpagCJhbhr85hmh0c0itxoRlc-vQ/view?usp=sharing)
>
>
> As a note, while we agree that sensitivity might be an issue, this metric allows us to investigate potential sources of biases. Therefore, we believe it is safer to be on the side of more false positives than false negatives, and (partial) confusion matrices can be explored to ensure the bias is not concerning as defined by different stakeholders.

---

> ### Author Response · Authors · 2023-11-23
> **Response (continued)**
>
> **While the paper introduces the SkewSize metric, it could potentially be replaced by just a chi-square test, offering a p-value to convey the significance of skewness.**
>
> As a reminder, SkewSize includes two elements: the effect size per class, and the aggregation across classes. It is important to note that this separation in two steps is *necessary* to capture biases in the distribution of model errors. This is easily exemplified by examining confusion matrices. Let’s assume a full confusion matrix with 3 classes, where the model confuses class 0 for class 2 in group A, and class 0 for class 1 in group B:
>
> Group A:
> |             |  Pred=0 |  Pred=1 | Pred=2 |
> |:-----------:|:------:|:------:|:------:|
> |   True=0  |  50 |  0 | 50 |
> |   True=1  | 0 | 50 | 0 |
> |   True=2  | 0 | 50 | 50 |
>
> Group B:
> |             |  Pred=0 |  Pred=1 | Pred=2 |
> |:-----------:|:------:|:------:|:------:|
> |   True=0  |  50 | 50 | 0 |
> |   True=1  | 0 | 50 | 50 |
> |   True=2  | 0 | 0 | 50 |
>
> Accuracy based metrics only focus on the diagonal terms, and would not highlight a bias. Metrics of the demographic parity type (i.e. ‘independence’ metrics as per [1]) would add all the rows before assessing the differences between groups. This is similar to the test proposed by Reviewer 6Zhw, which would not highlight bias. Similarly, equalized odds ('separation' metric as per [1]) computed on class 0 would not highlight bias, and the full confusion matrix is required to surface differences between the 2 groups.
>
> Therefore, while we agree that other tests could be considered (and have added text accordingly), we argue that SkewSize is currently the only metric that can surface biases in the distribution of model errors in an open-ended vocabulary. Our work further displays that these biases arise in vision language models, and that instruction tuning reduces their effect. We believe that this is an important message for the ICLR audience.
>
>
> [1] FAIRNESS AND MACHINE LEARNING: Limitations and Opportunities. Solon Barocas, Moritz Hardt, Arvind Narayanan. MIT Press, 2023.

---

### Official Review · Reviewer_6Zhw · 2023-11-01

**Soundness:** 4 excellent
**Presentation:** 3 good
**Contribution:** 2 fair
**Rating:** 5
**Confidence:** 4

**Summary:**

The paper presents a novel statistical measure- named SKEWSIZE, to qualify and detect bias present in a deep learning model. The proposed metric is shown to be more sensitive and exact in finding presence of bias, as validated by experiments across diverse datasets in different settings.

**Strengths:**

1. The presentation, writing and problem statement are clear.
2. Sheds light on utility of statistical measures in identification of bias in deep models.
3. Through three different settings, the authors demonstrate that the proposed SKEWSIZE metric successfully captures presence of bias more accurately than other conventional measures, for example, accuracy gap, worst group accuracy etc.

**Weaknesses:**

1. The comparison with respect to other benchmark methods is somewhat unfair. The authors should study effectiveness of statistical measures (in general) in identifying bias. For example, one could perform a statistical test (e.g. MWU test) with null hypothesis that the model scores for a subgroup (that represents a bias attribute) comes from the same distribution of as model scores without any subgrouping. An acceptance of the null hypothesis (using p-value) would indicate an absence of bias and rejection of the null hypothesis in favour of alternate hypothesis would indicate the presence of bias with respect to the subgrouping attribute.
2. Essentially, it is probably important to present benchmark comparisons on how the standard statistical measures and methods perform in identifying bias as baselines. Then only it presents us opportunity to appreciate the proposed specific statistical measure in bias identification.

**Questions:**

Refer to the weaknesses

---

> ### Author Response · Authors · 2023-11-16
> **Rebuttal**
>
> Thank you for taking the time to review our paper. We appreciate your constructive feedback and below we address the points raised in your review:
>
> *1- “The comparison with respect to other benchmark methods is somewhat unfair. For example, one could perform a statistical test (e.g. MWU test) with null hypothesis that the model scores for a subgroup (that represents a bias attribute) comes from the same distribution of as model scores without any subgrouping.”*
>
> We thank the reviewer for this suggestion. While there might be other settings (e.g. domain adaptation) where drawing inspiration from the MWU test is suitable, in the setting we consider (e.g. distributional bias), we believe it could give spurious results due to its sensitivity to model specific properties that are reflected in scores (e.g. model confidence). Also, using the scores is a less general method as it requires access to the raw model output; it is not usable when one only has access to the predictions of a model.
>
> In order to reflect the reviewer’s comment in our work, we have added to the discussion the following statement:
>
> “Aspects to be investigated in future work include how our work could be extended to continuous attributes by using a different statistic (e.g. Tramer et al., 2017; Brown et al., 2023).  While the selected aggregation approach and metrics are suitable for the cases we consider in this work, in this case, other testing statistics or aggregations might be more appropriate”.
>
> *2. “Essentially, it is probably important to present benchmark comparisons on how the standard statistical measures and methods perform in identifying bias as baselines. Then only it presents us opportunity to appreciate the proposed specific statistical measure in bias identification.”*
>
> We agree that other statistical methods could be considered in our evaluation in addition to the 3 accuracy based metrics and 2 fairness metrics we already considered in the paper. Following the reviewer’s suggestion, we have added to the manuscript (Appendix B) results with the Phi coefficient as a measure of effect size when computing SkewSize in the dSprites experiments. The results show that in this case the Phi Coefficient yields similar trends as the Cramer’s V. Notice, however, that it is not advisable to use the Phi Coefficient on contingency tables larger than 2x2, that is the reason why we decided to use the more general Cramer’s V when computing SkewSize throughout our work.
>
> |          | Cramer’s V - Class 0 | Cramer’s V - Class 1 | Cramer’s V - Class 2 | Phi - Class 0 | Phi - Class 1 | Phi - Class 2 |
> |:--------:|:--------------------:|:--------------------:|:--------------------:|:-------------:|:-------------:|:-------------:|
> | Unbiased |         0.012        |         0.011        |         0.019        |     0.012     |     0.011     |     0.027     |
> |  Class 0 |         0.670        |         0.015        |         0.016        |     0.948     |     0.015     |      0220     |
> |  Class 1 |         0.014        |         0.683        |         0.108        |     0.014     |     0.966     |     0.152     |
> |  Class 2 |         0.047        |         0.006        |         0.696        |     0.067     |     0.006     |     0.985     |
>
>
> We would greatly appreciate it if you let us know in case there are any other questions so we can further improve the manuscript based on your feedback.

---

> ### Author Response · Authors · 2023-11-20
> **Gentle nudge**
>
> Dear Reviewer 6Zwh,
>
> Thank you for your work in reviewing our paper. We have responded to your review, performed new experiments in light of your comments, and revised our paper to reflect your feedback. As the discussion period is approaching its end, we would appreciate it if you could go through our response, consider revising your review/score accordingly or let us know in case you have further questions.
>
> Kind regards,
> Authors

---

> > ### Comment · Reviewer_6Zhw · 2023-11-22
> >
> > Thank you for your response and experiments on Phi coefficient. I appreciate that.
> > However, it remains unclear to me why you say that the statistical test on scores is not right because of involvement of model specific properties? In the end, we are assessing the biasness of the model itself right? Am I missing something here, can you provide me some references based on your claim? Moreover, I also don't agree that often we don't have access to raw output of models. I would rather argue that, many a times, the architectural details, kind of model architectural information etc. may not be available, but we can fairly assume the availability of model's final output (in most cases).

---

> > > ### Author Response · Authors · 2023-11-22
> > >
> > > Dear Reviewer,
> > >
> > > Thank you for your follow-up questions.
> > >
> > > Apologies if there was any confusion in our prior response.
> > >
> > > **it remains unclear to me why you say that the statistical test on scores is not right because of involvement of model specific properties? In the end, we are assessing the biasness of the model itself right?**
> > >
> > > The test you propose, independently of the statistic used, is indeed valid. Different statistics can be used to provide a single p-value on the effect of the attribute on the predictions (as has been proposed in e.g. Tramer et al., 2017). This would correspond to an assessment of 'demographic parity', i.e. $f(X) \perp Z$ (which is one of our baselines), and would tell us whether the model is biased or not. This measure would likely be influenced by the model's calibration, e.g. if the model is over-confident for a specific group compared to others. This is a limitation of the demographic parity type of metrics, not of your suggestion in particular. As we describe in section 2.3, the test you propose can be performed using the effect size (rather than a p-value). We believe that effect size is more informative than a single p-value, but that choice needs to be application-driven.
> > >
> > > In our formulation, we are specifically interested in $f(X) \perp Z | Y $, i.e. equalized odds types of metrics, to better pinpoint the sources of the bias in a multiclass task. As you proposed, we could obtain a p-value per class. How to aggregate these p-values across classes is then not straightforward, and we selected to refer to the skewness for this end. As you mentioned, this is not the only possibility, and we have added text to our discussion accordingly (see response above). To make this clearer, we will amend the text in our discussion:
> > >
> > > *"Aspects to be investigated in future work include how our work could be extended to continuous attributes by using a different statistic (Brown et al., 2023). While the selected aggregation approach and metrics are suitable for the cases we
> > > consider in this work, other testing statistics or aggregations might be more appropriate based on the mathematical properties of the label and attributes, but also on fairness desiderata (e.g. demographic parity versus equalized odds). We hope our work contributes to the adoption of statistical methods to assess distributional biases in machine learning models."*
> > >
> > > **Moreover, I also don't agree that often we don't have access to raw output of models.**
> > >
> > > Apologies, we believed you meant to test the model representation (e.g. the penultimate layer output) rather than the model predictions. We agree with you that tests on model predictions are straightforward.
> > >
> > > We hope that these clarifications help you understand the value of our contribution, and how it can flexibly be adapted to other bias testing. We'd be happy to answer any further questions, and would greatly appreciate more feedback and/or a reconsideration of your score.

---

> > > > ### Author Response · Authors · 2023-11-23
> > > >
> > > > Dear Reviewer,
> > > >
> > > > We further describe the difference between demographic parity and equalized odds types of metrics in our latest response to Reviewer 1DWK, and display simple confusion matrices with biases that would not be captured by a test on predictions only.
> > > >
> > > > We hope this is helpful for your decision and thank you for your time and efforts in reviewing our work.

---

### Meta-Review · Area_Chair_8pBN · 2023-12-05

**Metareview:**

The authors propose SkewSize - a combination of existing statistical measures (more precisely, a ratio of the skewness to the variance suitably normalized) that can be used to identify certain skews in the error distribution (such as from spurious correlations). The authors demonstrate success on standard benchmarks and a VLM.

Strength: the authors set up the problem cleanly in terms of a set of effect size computations (via Cramer's V), and have a nice set of evaluations with standard baselines, including findings comparing the metric to measures such as worst-group accuracy, as well as experiments on gender occupation biases.

Weaknesses: the reviewers generally found that as a methodology paper, the novelty of the paper was weak (as it's a normalized ratio of existing effect size measures). As an empirical work, there were concerns about the overall rigor and soundness of the evaluation, including comparison to other statistics you could compute of the effect size or the lack of a 'ground truth' bias to compare the outcomes to.

The reviewers were split on the work, but most of the reviewers are borderline or negative, and the one positive review is very brief and low confidence.

**Justification For Why Not Higher Score:**

As a pure methods paper, the method itself is a somewhat ad-hoc combination of existing statistics
As an empirical paper, the paper would need to really do quite a bit morre to show that the method can e.g. identify many known dataset skews or help with model debugging.

**Justification For Why Not Lower Score:**

N/A

---

### Decision · Program_Chairs · 2024-01-16

Reject